# Measured and Perceived Effects of Upper Limb Home-Based Exergaming Interventions on Activity after Stroke: A Systematic Review and Meta-Analysis

**DOI:** 10.3390/ijerph19159112

**Published:** 2022-07-26

**Authors:** Axelle Gelineau, Anaick Perrochon, Louise Robin, Jean-Christophe Daviet, Stéphane Mandigout

**Affiliations:** 1HAVAE Laboratory UR 20217, University of Limoges, 87000 Limoges, France; anaick.perrochon@unilim.fr (A.P.); louise.robin@unilim.fr (L.R.); jean-christophe.daviet@unilim.fr (J.-C.D.); stephane.mandigout@unilim.fr (S.M.); 2Department of Physical Medicine and Rehabilitation, University Hospital Center, 87000 Limoges, France

**Keywords:** stroke, upper limb, exergames, activity, tele-health, e-health, home

## Abstract

After discharge from the hospital to home, stroke patients may experience weakness and reduced movement in their hemiparetic arms that limits their ability to perform daily activities. Therapists can use exercise games (exergames) to maintain functional abilities and daily use of the arm at home. A systematic review and meta-analysis was conducted to determine the efficiency of upper limb home-based rehabilitation, using exergaming on activity abilities in stroke. Randomized controlled trials were reviewed in the CENTRAL, MEDLINE, CINAHL, EMBASE, and SCOPUS online databases. Clinical measures of observation and self-reporting were studied in post-intervention and follow-up. Nine studies were included in this systematic review (535 participants). The Physiotherapy Evidence Database (PEDro) score was 6.6/10 (SD 1.0, range 5–8), indicating good quality. This systematic review and meta-analysis showed that upper limb home-based exergaming interventions were no more effective in terms of activity than conventional therapy after stroke, according to the observational and subjective assessments in post-intervention and follow-up. Using this same approach, future studies should focus on evaluating home-based exergames through subgroup analysis to be able to propose recommendations.

## 1. Introduction

Home-based rehabilitation is an ongoing rehabilitation approach to restore patients’ health in their home. Tele-rehabilitation refers to delivering rehabilitation and habilitation services via Information and Communication Technologies (ICTs) [1,2]. The challenge is to ensure the continuity of care at home for patients after the hospitalization period. A recent example was during the COVID-19 pandemic, as patients had difficulty obtaining their rehabilitation care either in a center or at home [3]. This disruption of care may lead to increased disability and morbidity, due to the lack of necessary rehabilitation care for people with continuing care needs [3,4]. The COVID-19 pandemic demonstrated the need to develop effective home-based strategies to assess and treat patients, support families, and train professionals in rehabilitation. Home-based rehabilitation can provide a solution to allow post-stroke patients to restore functional abilities in their home [5,6,7]. The systematic review by Chen et al. (2018) revealed that home-based rehabilitation technologies offer multiple benefits, such as improving motor skills, providing a quality of rehabilitation equivalent to conventional therapies, and improving patients’ daily life [7]. In addition to restoring functional abilities, tele-rehabilitation could also be used for patient follow-up [8].

Exercise games, or exergames, provide a supportive tool for home-based rehabilitation. According to the systematic review of Mat Rosly et al. (2017), exergaming is defined as the integration of physical activity into a video game environment that requires active body movements to control the game [9]. A close collaboration between the game developers, human movement scientists, and therapists is needed to design attractive games that implement therapeutic exercises for post-stroke, home-based rehabilitation [10]. Perrochon et al. (2019) reviewed the effect of home-based exercise games on the upper and lower limbs in individuals with neurological diseases, including stroke [11]. They indicated that these interventions constitute a relevant alternative for rehabilitation at home. These interventions were at least equivalent in effectiveness to conventional therapy or usual care [11]. The limitations were mainly due to the heterogeneity between the trials concerning population type, study design, interventions, and outcome measures. We cannot draw any conclusions specifically on the effects of home exergames on the upper limb (UL) for stroke patients. It would be interesting to focus on this approach.

Most of the stroke patients suffer from motor disorders, mainly affecting the functioning of the arm and hand [12]. Sensorimotor impairment of the paretic ULs negatively impacts on performance in the activities of daily living (ADLs) following stroke [13,14]. Residual sequelae in the UL may persist in affecting hand function and independence in ADLs [15]. Severity and ADL dependency may be associated with a decrease in the number of hours of activity of the affected UL. A decrease in the activity ratios, therefore, should be considered when designing therapeutic interventions and setting goals [16]. In recent literature, the intensity and frequency of exergaming recommended for subacute post-stroke patients was studied [17]. The authors reported that twice-daily exergaming, compared to high-intensity once-daily exergaming or lower-intensity once-daily standard care, improved the clinical and motor symptoms and the quality of life in the participants with subacute post-stroke [17]. These results are promising; however, they may not be transferable to a home setting that uses remote monitoring and commercially available technology. According to the systematic review of Schneider et al. (2016), increasing the amount of rehabilitation after stroke improves activity, but a large amount of extra rehabilitation needs to be provided to achieve a beneficial effect [18]. An indicator of effective home exergaming rehabilitation would be restoring the functional abilities of the post-stroke patient.

A recent meta-analysis by Wong et al. (2020) reported that home-based exercise involving technology and assistive devices (e.g., finger tracking, virtual reality, gaming, robotics, orthosis) was no more effective than home-based exercise without such devices in improving the post-stroke UL activity [19]. The authors did not exclusively assess the effects of exergames, but focused on self-administered, home-based UL exercise in improving post-stroke activity. In addition, most of the studies included in this systematic review and meta-analysis assessed the activities by observation (scales with standardized rating by a therapist) and only one by subjective assessments (i.e., self-report questionnaire) [19]. However, in upper limb rehabilitation after stroke, there may be inconsistencies between the actual ability assessed by therapists and the patients’ self-perception, including an over- or underestimation of their capacity [20]. Combining the self-reported measures and observational measures could help design rehabilitation strategies [21]. It would be interesting to further investigate the results of Wong et al. (2020), by focusing on studies with home-based exergames, that compare the measured and perceived effects in post-stroke UL activity.

Finding ways to help stroke patients exercise at home is crucial for their long-term and ongoing rehabilitation. The needs of people with stroke may remain unmet in the long term [22]. We wondered whether any post-intervention effects on activities would last over the long term. As reported by Bai et al. (2020), rehabilitation strategies, such as virtual reality, mirror therapy, or brain-computer interface (BCI) technology, may not have long-term effects [23]. Therefore, we investigated the immediate and the long-term impact of home-based exergaming rehabilitation of the upper limb activity after stroke.

The objective of this systematic review and meta-analysis was to examine the measured and perceived effects of UL home-based exergaming interventions on activity after stroke, compared with conventional therapy, in post-intervention and follow-up.

## 2. Materials and Methods

### 2.1. Search Strategy

We conducted this systematic review and meta-analysis according to the Preferred Reporting Items for Systematic Reviews and Meta-Analyses 2020 (PRISMA 2020) guidelines (Appendix A) [24].

We identified the most relevant articles in the Cochrane Central Register of Controlled Trials (CENTRAL; Cochrane Library), MEDLINE (PubMed search engine), Cumulative Index to Nursing and Allied Health Literature (CINAHL), EMBASE, and SCOPUS online databases. A combination of medical sub-headings (MeSH) and keywords was used to search each database (Appendix B).

The eligibility criteria are based on the PICOS elements of the review question [25]. The criteria were as follows: P, Post-stroke adults; I, Intervention using an exergaming technology for UL rehabilitation at home; C, Comparison with conventional therapy (i.e., usual practice); O, Outcome measures used to assess activities by observation and self-reporting; and S, Study design included only Randomized Controlled Trials (RCTs), the gold standard for evaluating the effectiveness of interventions. The article had to be in English and published before July 2021 to be included. The exclusion criteria were: articles published in other languages; non-RCTs; articles not reporting outcome measures; discussion/position papers or comments; conference papers; abstracts or articles without enough information about the intervention (study protocol); articles only reporting the development of the technology (study design); articles reporting feasibility; and preliminary results (pilot studies).

### 2.2. Selection of Studies

Two authors (AG, LR) screened independently the title and abstract of all of the search results to identify suitable studies, and then assessed all of the trials for eligibility, based on the full text. Disagreements between authors were resolved by consensus.

### 2.3. Data Extraction and Management

Data extraction was performed based on the current literature in the field and on the research questions. A pre-tested data collection and a final extraction were independently conducted by two authors (AG, AP). We collected data on the authors’ names, year of publication, design study, participants’ characteristics, technology rehabilitation and usual rehabilitation, intervention details (number of sessions, frequency, and length), outcome measures, and major findings. Disagreements regarding the data extraction were resolved by discussion or, if necessary, with a third author (SM). We contacted the study authors for additional information when necessary.

### 2.4. Quality Assessment

The RCTs were assessed using the Physiotherapy Evidence Database (PEDro) Scale, which generates a score out of 10 points, reflecting a study’s internal validity, the methodological quality, and the potential risk of bias in the study [26]. The PEDro results were interpreted using Foley’s quality assessment, where studies are rated as excellent to poor based on the following classification: a PEDro score of 9–10 is excellent; a score of 6–8 is good; a score of 4–5 is fair; and a score less than 4 is poor [27]. The scoring was completed by two of the authors (AG, SM). Any disagreements were resolved by consensus.

### 2.5. Outcome Measurements

The UL activity outcomes were used in the analysis according to the World Health Organization’s International Classification of Functioning, Disability and Health (ICF-WHO) framework [28]. They were classified into two categories, according to their mode of administration. First, there were observational assessments of the UL activity (by a therapist using a measuring scale with a standardized evaluation). We considered the Action Research Arm Test (ARAT), the Chedoke Arm and Hand Activity Inventory (CAHAI), the Barthel Index (BI), the Wolf Motor Function Test (WFMT), the Box and Block Test (BBT), the grooved pegboard, and the Nine Hole Peg Test (NHPT). Second, there were subjective assessments of the UL activity (through self-reporting by the patient). We considered the Motor Activity Log (MAL), the Canadian Occupational Performance Measure (COPM), the ABILHAND, and the Nottingham Extended Activities of Daily Living (NEADL).

### 2.6. Meta-Analysis

The meta-analysis was conducted using Review Manager (RevMan, version 5.4, The Cochrane Collaboration, London, UK, 2020). All of the relevant data, measured using different scales, were converted to a single scale. The primary outcome was taken as a priority, followed by the scales reported by most of the authors. Objective and subjective assessments were chosen in post-intervention and in follow-up. For each study, the absolute score (mean and standard deviation; SD) was recorded at the end of the treatment for the interventional and control groups. When the SD of the mean was not available, we requested it from the corresponding authors. The pooled results were estimated by calculating the standardized mean difference (SMD with 95% confidence intervals (CI)), when the studies measured UL activity, but used different psychometric scales. The mean difference (MD) with 95% confidence intervals (CI) was calculated when the studies used the same psychometric scale. The SMD method does not correct for differences in the direction of the scale; when some scales decreased with better performance, the mean values were multiplied by −1 to ensure that all of the scales were in the same direction [29]. The heterogeneity of the results was reported as I2 followed by a percentage: an I2 between 0–40% corresponds to low heterogeneity; between 30–60% corresponds to moderate heterogeneity; between 50–90% corresponds to substantial heterogeneity; and between 75–100% corresponds to considerable heterogeneity [29]. An I2 > 50% was considered heterogeneous. Forest plot graphics were created to present the pooled effect. The results were considered statistically significant when *p* < 0.05 in the equivalent z test.

## 3. Results

### 3.1. Study Identification

We identified 587 articles using the search strategy. A total of 468 records were screened, and 20 studies were eligible for inclusion in this systematic review based on the full-text. Within the 20 studies screened, nine RCTs satisfied the inclusion criteria and were included in this systematic review, and eight could be included in the meta-analysis (Figure 1).

### 3.2. Participants

The Interventional Group (IG) and the Control Group’s (CG) demographic characteristics are recorded in Table 1. The sample size varied between 18 and 235 participants, and included 535 stroke patients. There were more male participants in the sample size (57.2%) than female participants. According to the elapsed time from stroke (i.e., recovery phase) [30], one study included participants in the early subacute phase (7 days–3 months) [31], two studies in the late subacute phase (3–6 months) [32,33], and six studies in the chronic phase (>6 months) [34,35,36,37,38,39]. According to the impairment baseline, variability was found in the assessment (Table 1).

### 3.3. Interventional Group

The characteristics of the IG are found in Table 1. Five studies used nonspecific video game systems (e.g., the Nintendo Wii™ [31,33,35], the Microsoft Xbox Kinect™ [34,36], the Sony PlayStation EyeToy™ [36]). Some of the video-game systems have been combined with a system specifically designed for rehabilitation (e.g., Rehabilitation Gaming System (RGS) with a pair of data gloves [34], Wiimotes™ with a virtual glove [33], and MusicGlove [39]). Others used a specific rehabilitation device: Hand Mentor Pro (HMP) [32]; 3D motion tracking system (Polhemus 3, Space Fastrack, Colchester, VT, USA) [37]; SCRIPT dynamic wrist and hand orthosis and SaeboMAS (Saebo Inc., Charlotte, NC, USA) [38]. The number of video games varied from one [34,39] to six [36]. Eight of the studies focused on rehabilitating the hemiparetic arm; and only one study was carried out on both arms [34]. The type of movement required varied between studies, with two studies requesting patients to perform exercises in a standing position [35,36]. Supervision was implemented for some of the studies (e.g., for each session via a tele-rehabilitation system VRRS.net^®^ (The Massachusetts Institute of Technology, Cambridge, MA, USA) [37], a visit once per week [38], daily contact in the first week (phone or SMS) [36], contact at least once a week [39], or weekly contacts (phone or e-mail) [32]).

### 3.4. Comparison

The nature of the CG differed across the studies (Table 1). One study completed conventional physiotherapy at the center [37], one did usual care [33], while the others did their rehabilitation independently at home, using different means. For example, two of the studies were based on the Graded Repetitive Arm Supplementary Program (GRASP) [31,36], one was based on the modified Constraint-induced Movement Therapy (mCIMT) [35], and two others used an exercise book/pamphlet [38,39].

### 3.5. Training Settings

The time spent on interventions varied across the studies, with the intensity ranging from 30 [38] to 180 [32,39] min/day, the frequency ranging from 3 [39] to 7 [31,33] days/week, and the duration from 2 [35] to 8 [32,33] weeks. Six studies had follow-up [31,34,35,36,37,38,39], varying from 4 [36,37,39] to 24 [31,35] weeks. Further details of the training modalities are available in Table 1.

### 3.6. Outcome Measures

The results showed several outcome measures used for the activity across the studies (Table 1). Six studies assessed the activities by both observation and subjective assessments. Two other studies assessed only with an observation-based scale, and one study with only a self-reported scale.

Eight studies had observation assessments of the activity in the primary outcome (the Action Research Arm Test (ARAT) for four studies [31,32,36,38], the Wolf Motor Function Test (WMFT) timed tasks for two studies [33,35], the Chedoke Arm and Hand Activity Inventory (CAHAI) [34], and the Box and Block Test (BBT) [39].

Seven studies focused on self-reported activity assessments, six of which used the Motor Activity Log (MAL), and one used the ABILHAND. The MAL is composed of two subscales, one on the amount of use (AOU) and the other on the quality of movement (QOM) of the affected limb, and measures the activity performance in real-life situations post-stroke. Note that McNulty et al. (2015) [35] used the QOM subscale of the MAL (MALQOM) as the primary outcome.

### 3.7. Immediate Effects on Upper Limb Activity

When analyzing the overall observed outcomes, the meta-analysis indicated that the IG did not show any significant differential effects compared with the CG (eight studies, SMD = −0.05; 95% CI = −0.24 to 0.14; I^2^ = 0%; P = 0.62; fixed-effects model) (Figure 2a).

When analyzing the overall self-reported outcomes, the analysis focused on the amount of use (AOU) (Figure 2b) and quality of movement (QOM) (Figure 2c), the two subscales of the MAL. When analyzing the MALAOU, the meta-analysis indicated that the IG did not show any significant differential effects compared with the CG (four studies, MD = 0.15; 95% CI = −0.30 to 0.59; I^2^ = 48%; P = 0.52; fixed-effects model) (Figure 2b). When analyzing the MALQOM, the meta-analysis indicated that the IG did not show any significant differential effects compared with the CG (five studies, MD = 0.29; 95% CI = −0.06 to 0.64; I^2^ = 40%; P = 0.1; fixed-effects model) (Figure 2c).

### 3.8. Long Term Effects on Upper Limb Activity

Seven studies conducted a follow up. The follow-up times were inconsistent, ranging from 4 weeks [36,37,39] to 24 weeks [31,35]. Zondervan et al. (2016) [39] had the particularity of having proposed a crossover study.

When analyzing the overall observed outcomes, the meta-analysis indicated that the IG did not show any significant differential effects compared with the CG (five studies, SMD = 0.02; 95% CI = −2.0 to 0.24; I^2^ = 0%; P = 0.86; fixed-effects model) (Figure 3a).

When analyzing the MALAOU, the meta-analysis indicated that the IG did not show any significant differential effects compared with the CG (three studies, MD = −0.34; 95% CI = −0.74 to 0.06; I^2^ = 0%; P = 0.09; fixed-effects model) (Figure 3b). When analyzing the MALQOM, the meta-analysis indicated that the IG did not show any significant differential effects compared with the CG (four studies, MD = −0.15; 95% CI = −0.49 to 0.19; I^2^ = 15%; P = 0.38; fixed-effects model) (Figure 3c).

### 3.9. Quality Assessment

The mean PEDro scale score of the selected studies was 6.6/10 (SD = 1.0, range 5–8), indicating good quality (Table 2). There were no low-quality studies. The eligibility criteria, random allocation, and between-group statistical comparisons were reported in all of the studies. Eight studies described allocation concealment and baseline comparability. Seven studies reported that the assessor was blinded and only one study reported that the therapist was blinded. The participants were not blinded in any of the studies, which is understandable given the intervention.

## 4. Discussion

The objective of this systematic review and meta-analysis was to examine the effects of UL exergames at home on the activity after stroke compared with conventional therapy, considering both observation-based and self-reported assessments in the post-intervention and follow-up. In total, nine RCTs were included in this systematic review, one of which was excluded from the meta-analysis due to a lack of usable data.

### 4.1. Immediate Effects on Upper Limb Activity

Our meta-analysis found that the interventions based on exergames did not show significant superiority in the short term. Self-reported subjectivity was an additional standardized measure to the objective assessments, for confirming the findings. However, as exergames were not inferior to the controls, the results were considered as positive. Exergames, therefore, can be used as a support to tele-rehabilitation.

Comparing the studies was challenging, due to the heterogeneity of the interventional protocols (e.g., device, type of tasks, movement patterns, and training intensity). The intervention modalities varied in total intervention time, length, and frequency. However, the optimum frequency of intervention to achieve satisfactory results remains unclear [40]. Among the studies, there were both off-the-shelf recreational systems and systems specifically designed for rehabilitation at home. The few studies included in this systematic review and meta-analysis did not allow for a comparison of the two types, as executed by Maier et al. (2019) with virtual reality devices [41]. These authors found that specific systems showed a higher impact on recovery of activity than the control group, while the non-specific systems did not. Several general principles support the post-stroke rehabilitation process, and some have been studied using virtual reality (VR). Maier et al. (2019) reported that specific VR systems could potentially lead to a greater effect on recovery: task-specific exercises, explicit feedback, increasing difficulty, implicit feedback, variable practice, and mechanisms that promote the use of the paretic limb [41].

Only three studies included patients in the post-stroke sub-acute phase, which did not allow the subgroup to be studied. Wong et al. (2020) reported no significant differences in upper limb activity for the sub-acute and chronic populations when comparing the results of self-administered home-based assessments to no intervention [19]. However, in the literature, we found a discrepancy between the self-reported and objective assessments of arm function, especially in the acute and subacute phases [35]. Therapists offering home-based exergaming cannot determine which patients are most likely to benefit from the tool or experience side effects; further RCTs are needed to determine the profile of the patients responding to home-based exergaming therapy. Valid predictors of the response to therapy would be useful, to be able to select the appropriate patients for this specific therapy [42,43]. Moreover, five studies provided supervision during their intervention, although whether supervision affects ADLs needs to be determined. Regular supervision at home with monitoring of actual physical activity may encourage practice. As suggested by Mandigout et al. (2021), evaluating the effect of an individualized coaching program, the use of three modalities (activity tracker, phone calls, and visits) would facilitate the retention of physical activity levels [36]. Allowing the patient to work independently with the exergames would free up therapists’ time for therapeutic education. A recent study showed that self-managed motor-gaming with behavioral telehealth visits had outcomes to the same degree as in-clinic Constraint-Induced Movement therapy [44]. The authors indicated that the time saved could be redirected to behavioral interventions to improve the involvement of the paretic arm in daily life. These results would make it possible to rethink the support provided to post-stroke patients at home.

### 4.2. Long Term Effects on Upper Limb Activity

Seven studies included post-intervention follow-up at various time points, ranging from four weeks to 24 weeks. The pooled effect size showed no significant difference between the groups. Therefore, the current review showed that the long-term effects of exergaming training were not evident. Nevertheless, the retention of the benefits from these interventions should be cautiously interpreted. The intervention follow-ups were generally short, not allowing for a long-term assessment of the effects on daily arm activity.

Recently, a six-month retention of MAL and WMFT gains was found by Gautier et al. (2021) [44]. The authors suggested that “tune-up” sessions may be required to sustain MAL improvements in the long term. As in the study of Brouwer et al. (2018), assessing the effectiveness of client-centered “Tune-Ups” on community reintegration, mobility, and quality of life after stroke [45], this could similarly be proposed to focus on UL activities. As has been assessed in older adults for fall prevention, periodic interventions with exergaming over several years could be tested to determine the adherence rate and the impact on the activity level of post-stroke patients [46].

While the clinical-rated or self-reported scales have been clinically validated, they are mostly retrospective and susceptible to reporting bias and error [47]. To provide an objective and clinically valid measurement of an individual’s ADLs over the long term, real-time motion sensors could be used in the patient’s environment. Measuring the ADLs using wearable technologies could complement the current clinical assessments of post-stroke patients [47].

### 4.3. Risk of Bias

To maximize the quality of evidence in this review, all of the included studies were RCTs. As evaluated formally using the PEDro scale, the overall methodological quality of the included studies was good. Two of them were judged with a score of eight [31,32]. No studies were of excellent quality. Not surprisingly, patients and therapists were not blinded in this type of intervention. However, the assessors could be blinded, which was not the case for all of the studies.

According to the Cochrane Collaboration, the tests for funnel plot asymmetry should be used only when at least 10 studies are included in the meta-analysis; therefore, publication bias could not be tested.

### 4.4. Limitations

The limitations should be considered when determining the effect of home-based exergaming therapies on UL activity. The first limitation was the low number of trials included in the review: a sufficient sample size would allow the effects to be measured with a reduced potential error. Moreover, our analyses did not allow for comparisons of the subgroups, including stroke stages and initial severity level. The inclusion of more studies would determine whether the effects differed in the post-stroke acute, sub-acute, or chronic phases. We could have compared our results with the meta-analysis of Mekbib et al. (2020), to see if they were consistent with the knowledge of neuroplasticity-induced motor recovery [48]. The stratification of participants according to the initial level of motor and cognitive impairment would have allowed a better approach to the effects. A difference in the patients’ self-analysis, according to their initial motor and cognitive severity level, could have influenced their judgment of their own functional performance. These aspects could be compared with sufficient future data. Robust conclusions could not be drawn, due to the lack of standardized outcome activity measurements, highlighting the need for a gold standard in future research in this area.

### 4.5. Perspectives

This systematic review and meta-analysis suggest the combined use of observation and self-report assessments for measuring the effects of upper limb exergames at home on activity after stroke., Assessment complementarity is essential in routine clinical practice, research for future studies, and systematic reviews to provide a global view of the patients’ functional abilities. Ng et al. (2008) reported that a functional observational assessment, such as the WMFT, provided a comprehensive evaluation of the patient’s abilities, followed by an assessment, such as the ARAT, to focus on grasping, gripping, and pinching. The MAL assessment for additional qualitative information helps to understand the patient’s use of the hand in daily life [49].

Patient activity must be included in the overall care. An assessment of the patient outcomes, such as the effectiveness of motor function, is now well established in clinical research [50]. As reported in the Dalmazane et al. (2021) systematic review on multiple sclerosis, the consideration of fatigue and cognitive function would be relevant [51]. Stroke survivors may experience fatigue that can affect the motivation for rehabilitation and return to activity. In addition, the link between cognitive impairment and activity limitations and participation restrictions has been identified in the literature [52]. It would be wise to explore this approach for future studies on the possible benefits of home-based exergaming. Little is known about how the quality of life, satisfaction, and perceived environmental quality differ, according to the activity level. These perspectives would suggest looking beyond the activity to the participation components when assessing the daily experience of living at home with stroke. Additional factors (psychosocial, cognitive, and environmental) could influence participation [53]. Consideration of these factors may improve home rehabilitation programs by ensuring continuity, follow up, and the participation of individuals and families.

## 5. Conclusions

Exergames in upper limb tele-rehabilitation seem to be effective in promoting post-stroke activity. Some of the situations may be favorable for remote rehabilitation interventions. Exergames should be included in the global tele-rehabilitation to ensure continuity, follow-up, and the participation of individuals and families. The evaluation of the observational and self-reported measures detected convergences in the assessment of upper limb function after stroke. It would be relevant to consider both types of assessment in future studies to ensure the same results. The complementarity of assessments is essential in clinical routine practice and research, and for future studies and systematic reviews, in considering activity outcome measures in their entirety. Patient care should consider functional capacity, self-reported performance, and actual day-to-day performance.

## Figures and Tables

**Figure 1 ijerph-19-09112-f001:**
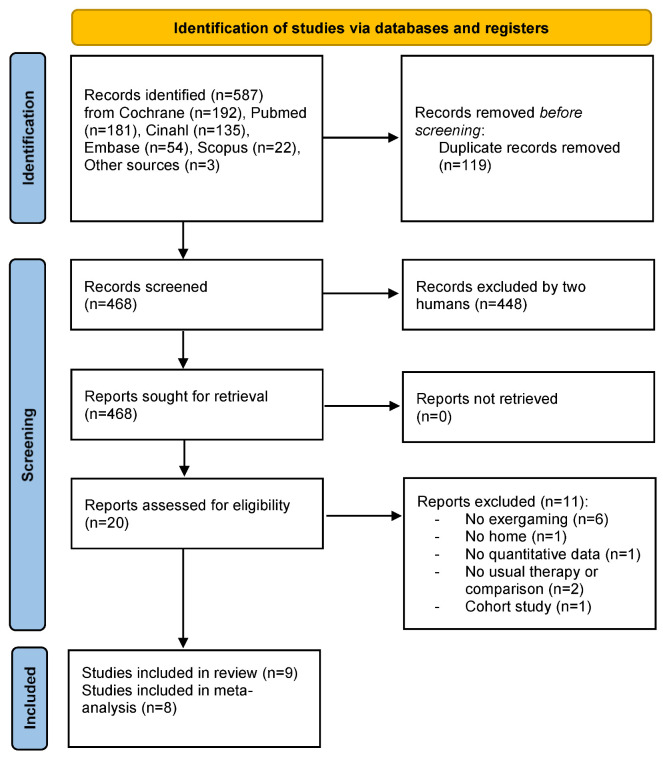
Prisma 2020 Flow Diagram [24].

**Figure 2 ijerph-19-09112-f002:**
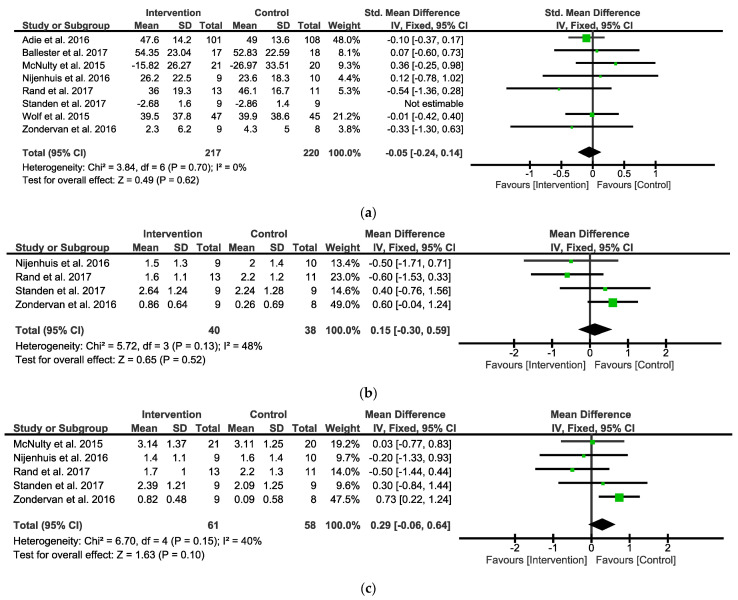
Forest plots between Intervention and Control Groups in post intervention. (**a**) SMD for the observed outcome; (**b**) MD for the self-reported outcomes MALAOU; (**c**) MD for the self-reported outcomes MALQOM. A pooled result favoring Intervention Group indicates negative values, and favoring Control group indicates positive differences between Intervention and Control Groups. Note: SMD = Standardized Mean Differences; MD = Mean Difference; MALAOU = Motor Activity Log Amount of Use; MALQOM = Motor Activity Log Quality Of Movement [31,32,33,34,35,36,38,39].

**Figure 3 ijerph-19-09112-f003:**
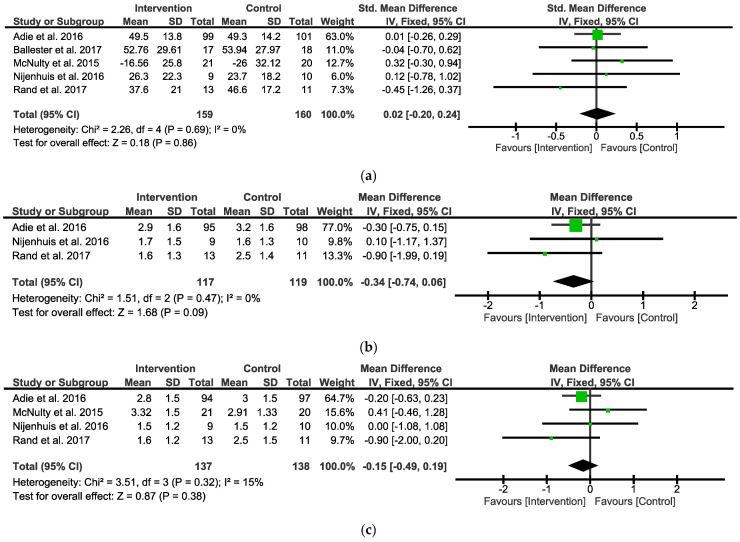
Forest plot between Intervention and Control Groups in follow-up. (**a**) SMD for the observed outcome; (**b**) MD for the self-reported outcomes MALAOU; (**c**) MD for the self-reported outcomes MALQOM. A pooled result favoring Intervention Group indicates negative values, and favoring Control group indicates positive differences between Intervention and Control Groups. Note: SMD = Standardized Mean Differences; MD = Mean Difference; MALAOU = Motor Activity Log Amount of Use; MALQOM = Motor Activity Log Quality Of Movement [31,34,35,36,38].

**Table 1 ijerph-19-09112-t001:** Characteristics of the included RCT studies (*n* = 9) (main outcome in bold).

Author (Year) [ref.]	Study Design Intervention Time Follow Up	Participants Sample Size *n*, Male/Female, Age Year, Months since Stroke Impairment (Min–Max) Mean (SD) or Median Mean (Interquartile Range)	Interventional Group (IG) Hardware Software Movements Control Group (CG) Type Movements	No of Sessions, Frequency and Length	Outcome Measures	Main Findings End of Intervention Effectiveness for IG Comparison with CG Follow Up Effectiveness for IG Comparison with CG	Risk of Bias (PEDro)
Adie et al. (2016) [31]	RCT, multicentric 6 wk. 24 wk.	235 subacute patients IG: *n* = 117, 66/51, 66.8 (14.6) yr., 1.9 (1.6) ARAT score (0–57): 41.2 (15.9) CG: *n* = 118, 65/53, 68.0 (11.9) yr., 1.8 (1.6) ARAT: 41.0 (16.6)	IG: Wii™ remote control 4 Wii™ sports games Arm unilateral movements CG: Arm exercises based on the GRASP n.a.	15 min warm-up, 45 min/d, 6 wk.	*Observation:* - **ARAT** *Self-report:* - MAL-14 - COPM	Both groups showed an improved arm function (ARAT) No significant difference in arm function (ARAT) between groups Both groups showed an improved arm function (ARAT) or occupational performance (COPM) No significant difference between groups	8
Ballester et al. (2017) [34]	RCT 3 wk. 12 wk.	35 chronic patients IG: *n* = 17, 8/9, 65.1 (10.3), 35.3 (25.2) CAHAI-13 (13–91): 52.8 (23.1) CG: *n* = 18, 6/12, 61.8 (12.9), 26.2 (13.9) CAHAI-13: 53.5 (22.5)	IG: RGS system—Kinect™ motion capture device, a pair of data gloves (DGTech Engineering Solutions) 1 custom game with 3 subtasks Shoulder, elbow and finger bilateral movements CG: horizontal and vertical stacking and unstacking of plastic cups Shoulder, elbow and finger bilateral movements	1–3 sessions, 5/wk., 3 wk. IG: 2min30 AEMF + 26min40 session (10 min per hand) CG: 20 min (10 min per hand)	*Observation:* - **CAHAI-13** - BI *Self-report:* n.a.	Significantly greater in arm function (CAHAI-13) for the IG Significant difference between groups in favor of IG No retention of the improvements in arm function (CAHAI-13) No significant difference between groups	6
McNulty et al. (2015) [35]	RCT 2 wk. 24 wk.	41 chronic patients IG: *n* = 21, 13/8, 59.9 (13.8), 11.0 (3.1) Motor classification (low:moderate:high): 3/8/10 CG: *n* = 20, 18/2, 56.1 (17.0), 6.5 (2.1) Motor classification: 5/7/8	IG: WMT—Wii™ remote control (+ self-adhesive wrap if poor grip strength) 5 Wii™ sports games + game specific drills Arm and hand unilateral movements, standing CG: mCIMT—A mitt worn on the less-affected hand, shaping practice tailored Arm and hand unilateral movements	60 min, 10 sessions, 2 wk. CG: Arm use 90% of the walking time + 15–20 min training tasks	*Observation:* - **WMFT-tt** - BBT - Grooved pegboard *Self-report:* - **MALQOM**	Both groups showed an improved arm function (WMFT-tt) and perceived daily use (MALQOM) No significant difference between groups Both groups showed maintained improvements in arm function (WMFT-tt) No significant difference between groups	7
Nijenhuis et al. (2016) [38]	RCT, pilot 6 wk. 8 wk.	20 chronic patients IG: *n* = 10, 7/2, 58 (48–65), 11 (10–26) ARAT score (0–57): 31.0 (3.5–50.0) CG: *n* = 10, 3/7, 62 (54–70), 12 (10–30) ARAT: 25.0 (3.8–30.8)	IG: SCRIPT (custom passive dynamic wrist and hand orthosis sensor), SaeboMAS (gravity compensation of the proximal arm) 3 custom games Arm and hand unilateral movements CG: Conventional exercises from an exercise book 34 exercises Arm and hand unilateral movements	30 min, 6/wk., 6 wk.	*Observation:* - **ARAT** - BBT *Self-report:* - MAL	No significant difference between groups IG showed moderate improvements for arm function (ARAT) No significant difference between groups	5
Piron et al. (2009) [37]	RCT 4 wk. 4 wk.	36 chronic patients IG: *n* = 18, 11/7, 66.0 (7.9), 14.7 (6.6) n/a CG: *n* = 18, 10/8, 64.4 (7.9), 11.9 (3.7) n/a	IG: VRRS.net^®^: 3D motion tracking system (Polhemus 3Space Fastrack^®^), a magnetic receiver attached to a real object, videoconferencing system 5 virtual tasks Arm unilateral movements CG: Conventional physiotherapy in the center Arm unilateral movements + postural control	60 min, 5/wk., 4 wk.	*Observation:* n.a. *Self-report:* - ABILHAND	Both groups showed significant improvements in patient’s perceived manual ability (ABILHAND) No significant difference between groups Both groups showed maintained improvements in patient’s perceived manual ability (ABILHAND) No significant difference between groups	6
Rand et al. (2016) [36]	RCT, pilot 5 wk. 4 wk.	24 chronic patients IG: *n* = 13, 9/4, 59.1 (10.5), 19.6 (11.3) FMA-UE (0–60): 35.4 (11.0) CG: *n* = 11, 6/5, 64.9 (6.9), 13.0 (6.0) FMA-UE: 41.3 (10.7)	IG: Kinect™ (sensor) and EyeToy™ (camera) 6 commercial games: 3 Kinect™ and 3 EyeToy™ games Shoulder and elbow unilateral and trunk movements, standing CG: 15–25 exercises and activities of the GRASP Arm and hand bilateral movements, seated	60 min, 6/wk., 5 wk.	*Observation:* - **ARAT** - BBT *Self-report:* - MAL	Both groups showed significant improvements in arm function (ARAT), perceived daily use (MAL) and manual dexterity (BBT) No significant difference between groups Both groups showed significant improvements in arm function (ARAT) No significant difference between groups	6
Standen et al. (2017) [33]	RCT 8 wk. None	27 subacute patients IG: *n* = 17, 8/9, 59 (12.03), 5.5 (4.0–14.9) Median WFMT (seconds): 2.60 (1.65, 6.00) CG: *n* = 10, 8/2, 63 (14.06), 4.0 (2.0–5.1) Median WFMT (seconds): 3.34 (1.9–4.9)	IG: Virtual glove with Wiimotes™ 3 custom games Arm and hand unilateral movements CG: Usual care n/a	IG: Max 20 min, 3/d, 8 wk. CG: n/a	*Observation:* - **WMFT** - NHPT *Self-report:* - MAL - NEADL	IG showed significantly greater change in perceived daily use (MAL) A significant difference between groups in favor of IG n.a.	6
Wolf et al. (2015) [32]	RCT, multicentric 8 wk. None	99 subacute patients IG: *n* = 51, 25/26, 59.1 (14.1), 3.8 (1.7) FMA-UE (0–66): 34.1 (12.1) CG: *n* = 48, 31/20, 54.7 (12.2), 4.2 (1.5) FMA-UE: 33.3 (12.0)	IG: HMP + HEP Robotic device with pneumatic artificial muscle, touchscreen, and Web-based monitoring (n/a) games HMP wrist and fingers unilateral movements + HEP shoulder/arm, elbow/forearm, wrist/hand, and task-based activities CG: classic home exercise program HEP shoulder/arm, elbow/forearm, wrist/hand, and task-based activities	180 min, 5/wk., 8 wk. IG: 120 min of robotic-based exercises and 60 min of functional-based activities CG: 120 min of traditional impairment-based exercises and 60 min of functional activities	*Observation:* - **ARAT** - WMFT *Self-report:* n.a.	Both groups showed significant improvement in arm function (ARAT and WMFT) A difference between groups were observed for CG on WMFT n.a.	8
Zondervan et al. (2016) [39]	RCT, crossover 3 wk. 4 wk.	18 chronic patients IG: *n* = 9, 5/3, 59 (35–74), 60.0 (48.0) FMA-UE (0–66): 56.4 (6.3) CG: *n* = 9, 5/4, 60 (45–74), 36.0 (12.0) FMA-UE: 53.8 (8.9)	IG: MusicGlove: an instrumented glove, laptop screen 1 custom game Fingers and thumb unilateral movements CG: a pamphlet of hand exercises Unilateral passive/active wrist, hand and fingers movements	180 min, 3/wk., 3 wk.	*Observation:* - **BBT** - NHPT - ARAT *Self-report:* - MAL	Both groups showed significant improvement in dexterity (BBT) No significant difference between groups IG showed significantly greater improvements in perceived daily use (MAL) A significant difference between groups in favor of IG	7

Note. RCT = randomized controlled trial; GRASP = Graded Repetitive Arm Supplementary Program; RGS = Rehabilitation Gaming System; AEMF = Automated Evaluation of Motor Function; WMT = Wii-based Movement Therapy; mCIMT = Modified-constraint therapy; FMA-UE = Fugl-Meyer Assessment for upper extremity; HMP = Hand Mentor Pro; HEP = Home exercise program; ARAT = Action Research Arm Test; WFMT = Wolf Motor Function Test; n/a = no answer; COPM = Canadian Occupational Performance Measure; CAHAI = Chedoke Arm and Hand Activity Inventory; MALQOM = Motor Activity Log Quality of Movement scale; NEADL = Nottingham Extended Activities of Daily Living; BBT = Box and Block Test; NHPT = Nine Hole Peg Test; IG = Interventional Group; CG = Control Group; ST = Short-Term; LT = Long-Term.

**Table 2 ijerph-19-09112-t002:** Quality assessment of selected randomized controlled trials using the Physiotherapy Evidence Database (PEDro) scale: a higher score implies improved quality.

Authors (Year)	C1	C2	C3	C4	C5	C6	C7	C8	C9	C10	C11	PEDro Score
Adie et al. (2016) [31]	YES	YES	YES	YES	NO	NO	YES	YES	YES	YES	YES	8
Ballester et al. (2017) [34]	YES	YES	YES	YES	NO	NO	NO	YES	NO	YES	YES	6
McNulty et al. (2015) [35]	YES	YES	YES	YES	NO	YES	YES	YES	NO	YES	NO	7
Nijenhuis et al. (2016) [38]	YES	YES	YES	YES	NO	NO	NO	YES	NO	YES	NO	5
Piron et al. (2009) [37]	YES	YES	YES	YES	NO	NO	YES	NO	NO	YES	YES	6
Rand et al. (2017) [36]	YES	YES	NO	YES	NO	NO	YES	YES	NO	YES	YES	6
Standen et al. (2017) [33]	YES	YES	YES	NO	NO	NO	YES	YES	NO	YES	YES	6
Wolf et al. (2015) [32]	YES	YES	YES	YES	NO	NO	YES	YES	YES	YES	YES	8
Zondervan et al. (2016) [39]	YES	YES	YES	YES	NO	NO	YES	YES	NO	YES	YES	7

Note: C1 = Eligibility criteria; C2 = Random allocation; C3 = Concealed allocation; C4 = Baseline comparability; C5 = Blinded subjects; C6 = Blinded therapists; C7 = Blinded assessors; C8 = Adequate follow-up (Drop-out rate); C9 = Intention-to-treat analysis; C10 = Between-group comparisons; C11 = Point estimates and variability.

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
