# Peer review of "Measured and Perceived Effects of Upper Limb Home-Based Exergaming Interventions on Activity after Stroke: A Systematic Review and Meta-Analysis"

_ijerph, 2022, doi:10.3390/ijerph19159112_

Round 1

Reviewer 1 Report

In this paper, the authors present a systematic review to determine the efficiency of upper limb home-based rehabilitation using exergaming on activity abilities in stroke.

The paper is well-written, the methods are well-described, the analyzed methods are comprehensive enough, the used statistical methods are appropriate, and the conclusions mostly aligned with the results. The only two main things that I would like the authors to improve are the following:

1. Revise the writing one more time (there are some minor English grammar mistakes and/or typos).

2. Revise every time a conclusion was extrapolated from what it should be. For example, extrapolating the paper's conclusions to tele-rehabilitation since moving to tele-rehabilitation involves a lot more details that were not analyzed.

Author Response

  1. [Revise the writing one more time (there are some minor English grammar mistakes and/or typos)]

Author’s response: Thank you for your advice. We have made some changes in the manuscript. Due to the tight deadlines for returning the manuscript, the editorial department we work with for English corrections was not able to return the text to us. But we can call on their service if there are still errors.

Action: see changes highlighted in the text. For example: in l.100 “crucial for their long-term”, l.120 “were”, l.141 “were”, l.174 “groups” l.240-242 “the”

  1. [Revise every time a conclusion was extrapolated from what it should be. For example, extrapolating the paper's conclusions to tele-rehabilitation since moving to tele-rehabilitation involves a lot more details that were not analyzed.]

Author’s response: Indeed, we have taken your comment into account. Then, we decided to use the word “support” instead of the word “alternative”. The exergame is one of the different modalities of telerehabilitation in the accompaniment around the person.

Action: see changes in conclusion highlighted

L531 “This systematic review found that exergames in upper limb tele-rehabilitation seemed to be a relevant support for promoting post-stroke activity. Some situations may be favorable for remote rehabilitation interventions. The exergames should be included in the overall tele-rehabilitation ensure continuity, follow-up and participation of individuals and families”.

Reviewer 2 Report

First of all, I would like to express my sincere thanks to the authors and editors of the journal for allowing me to review the manuscript entitled: "Measured and perceived effects of upper limb home-based exergaming interventions on activity after stroke: A systematic review and meta-analysis". In this sense, I would like to highlight the interest of the topic raised given that stroke-associated disability is a frequent complication and should be addressed as a major health problem. In turn, the use of new technologies as part of rehabilitation treatment should be studied in depth for its potential to minimise the impact of stroke on the patient, especially when it allows for a continuous rehabilitation process in the patient's own home.

However, I believe it is appropriate for the authors to make a number of clarifications to the information contained in the manuscript.

-       In the introduction to this study, the authors highlight the need to look for rehabilitation alternatives to conventional methods to minimise disability in stroke patients. However, they do not justify the relevance of the implementation of upper limb rehabilitation exercises. It would be pertinent to detail the reasons why it is important to carry out this type of rehabilitation, specifying the repercussions for the patient, his or her family and for the health system.

-       Regarding the methodology, it would be of great interest to detail the search strategy used, as this is a decisive element for other researchers to be able to replicate the search in case of interest.

-       How has the inclusion of studies involving patients at different times of the disease (in the period just after the stroke and others who suffered this pathology weeks ago) influenced the results? Please clarify this information in the manuscript. Similarly, the included studies have used different types of exercises and even different technological devices. How does this influence the results?

-       Finally, with regard to the discussion, it would be of interest to add what future lines of research could be based on the results obtained.

Author Response

  1. [Comment: In the introduction to this study, the authors highlight the need to look for rehabilitation alternatives to conventional methods to minimise disability in stroke patients. However, they do not justify the relevance of the implementation of upper limb rehabilitation exercises. It would be pertinent to detail the reasons why it is important to carry out this type of rehabilitation, specifying the repercussions for the patient, his or her family and for the health system.]

Author’s response: Thank you for the opportunity to detail the repercussions of stroke on the upper limb and the implementation of rehabilitation. It is relevant to consider all the actors around the patient, including the family and the professionals for an optimal collaboration.

Action: see changes highlighted

l.64-71: “Most stroke patients suffer from motor disorders, mainly affecting the arm and hand function [12]. Sensorimotor impairment of paretic ULs negatively impacts performance in activities of daily living (ADLs) following stroke [13,14].Residual sequelae in the UL may persist affecting hand function, activities and independolloent ADL [15]. Severity and dependence in ADLs may be associated with a decrease in the number of hours of activity of the affected UL and a decrease in activity ratios that should be considered when designing therapeutic interventions and setting goals [16].“

l37 “The Covid-19 pandemic has demonstrated the need to develop effective home-based strategies to assess and treat patients, support families and train professionals to continue rehabilitation.”

  1. [Comment: Regarding the methodology, it would be of great interest to detail the search strategy used, as this is a decisive element for other researchers to be able to replicate the search in case of interest.]

Author’s response: We agree with your point. We have considered this in section "2.1 Search Strategy" with Appendix B. detailing the search equation used for the search strategy. We have chosen to put the search equation in appendix so as not to overload the text. Thank you.

Appendix B. Search strategy

((((((stroke[title/abstract] or "post-stroke"[title/abstract] or poststroke[title/abstract] or hemipl*[title/abstract] or hemipar*[title/abstract] or "cerebrovascular disease"[title/abstract] or "brain injur*"[title/abstract])) and ("virtual reality"[title/abstract] or "vr gam*"[title/abstract] or "video gam*"[title/abstract] or "videogam*"[title/abstract] or "video-gam*"[title/abstract] or "computer gam*"[title/abstract] or "gaming system*"[title/abstract] or "serious gam*"[title/abstract] or "exercise gam*" or exergam*[title/abstract] or exer-gam*[title/abstract] or "commercial gam*"[title/abstract] or "rehabilitation gam*"[title/abstract] or "augmented reality"[title/abstract] or gamification[title/abstract] or "vr-based"[title/abstract] or "video-based"[title/abstract] or "computer-based"[title/abstract] or "sensor-based"[title/abstract] or Wii[title/abstract] or Nintendo[title/abstract] or "X-box"[title/abstract] or Kinect[title/abstract] or microsoft[title/abstract] or "play-station"[title/abstract] or playstation[title/abstract] or interactive[title/abstract] or "assistive gam*"[title/abstract] or neurogam*[title/abstract] or "gaming technolog*"[title/abstract] or technolog*[title/abstract] or "User-Computer Interface"[title/abstract])) and (rehabilitation[title/abstract] or neurorehabilitation[title/abstract] or telerehabilitation[title/abstract] or "tele-rehabilitation"[title/abstract] or telecare[title/abstract] or "rehab system*"[Title/Abstract] or rehab[Title/Abstract] or conventional[title/abstract] or physiotherapy[Title/Abstract] or "physical therapy"[Title/Abstract] or "occupational therapy"[Title/Abstract] or telemedicine[Title/Abstract])) and ("upper extremit*"[title/abstract] or "upper limb*"[title/abstract] or "upper-limb*"[title/abstract] or arm*[title/abstract] or hand*[title/abstract])) and (home[title/abstract] or "home-based"[title/abstract] or "in-home"[title/abstract] or domiciliary[Title/Abstract] or "self-directed"[Title/Abstract] or "self-administered"[Title/Abstract] or "self-practice"[Title/Abstract])) not (children[title/abstract] or child[title/abstract])

  1. [Comment: How has the inclusion of studies involving patients at different times of the disease (in the period just after the stroke and others who suffered this pathology weeks ago) influenced the results? Please clarify this information in the manuscript. Similarly, the included studies have used different types of exercises and even different technological devices. How does this influence the results?]

Author’s response: We added information to the discussion on these two points, thank you.

Action: see changes highlighted

l.417 “Several general principles underpin the process of stroke rehabilitation, and some have been studied in the context of virtual reality (VR). Maier et al (2019) reported that specific VR systems could potentially lead to a greater effect on recovery: task-specific practice, explicit feedback, increasing difficulty, implicit feedback, variable practice, and mechanisms that promote the use of the paretic limb [41]. “

l.423 “According to the post-stroke phase, only three studies included patients in the sub-acute phase, which did not allow for an adequate subgroup study. According to Wong et al. (2020), there is no significant difference on upper limb activity for sub-acute and chronic populations when comparing self-administered home-based to no intervention [19]. However, in the literature, we found a discrepancy between self-reported and objective assessments of arm function, especially in the acute and subacute phases [34]. Therapists offering home-based exergaming interventions may not know which people are most likely to benefit or experience side effects. Further RCTs are warranted to determine the profile of patients responding to home-based exergaming therapy. Valid predictors of response to therapy would be needed to select appropriate patients for this specific therapy [42,43]. “

  1. [Comment: Finally, with regard to the discussion, it would be of interest to add what future lines of research could be based on the results obtained]

Author’s response: in perspective, it seemed important to us to open up in the field of participation, which is still under-assessed today despite the fact that it is an ICF dimension.

Action: see changes in p. highlighted

l.522 “Little is known about how the quality of life, satisfaction, and perceived environment quality differ according to activity level. These perspectives would suggest looking beyond activity to participation components when assessing the daily experience of living at home with stroke. Additional factors (psychosocial, cognitive and environmental) could influence participation [53]. Consideration of these factors could lead to improved home rehabilitation programmes by ensuring continuity, follow-up and participation of individuals and families.”

Reviewer 3 Report

Dear authors,

Thank you for the opportunity to review your manuscript. It is a work that focuses on a topic of recent importance, especially in light of the pandemic we have fought.

26 Tele-health; E-health

29 It’s not alternate. “A continuative rehabilitation approach to discharge in the home setting”

43 Actually, several reviews have been compiled on the topic and on neurological disorders, also of a network nature, perhaps you should expand, perhaps adding these references: https://doi.org/10.1007/s10072-021-05855-2 , https://doi.org/10.1080/09638288.2020.1768301 , https://doi.org/10.1089/g4h.2012.0073 , https://doi.org/10.1016/j.ijmedinf.2018.12.001

L52 from the point of view of evidence you need high intensity and high frequency, more than hard work ..

57 “In addition to maintaining functional improvements, telerehabilitation could monitor the follow up”

122 Risk of Bias is missing

150 Did you only conduct MD or did you expect to use SMD? For what heterogeneity value did you conduct a random effect?

Figure 2 it’s non SMD .. it’s MD.. Is it the same outcome evaluated?

466 PROSPERO is missing

508 Again, I disagree on the term alternative .. there seems to be better and worse intervention, instead telerehabilitation can provide continuity, monitor and engage individuals and families.

Author Response

  1. [Comment: 26 Tele-health; E-health]

Author’s response: completed

Action: see changes in l.26 highlighted

  1. [Comment: 29 It’s not alternate. “A continuative rehabilitation approach to discharge in the home setting”]

Author’s response: completed

Action: see changes in l.29 highlighted

  1. [Comment: 43 Actually, several reviews have been compiled on the topic and on neurological disorders, also of a network nature, perhaps you should expand, perhaps adding these references: https://doi.org/10.1007/s10072-021-05855-2, https://doi.org/10.1080/09638288.2020.1768301, https://doi.org/10.1089/g4h.2012.0073, https://doi.org/10.1016/j.ijmedinf.2018.12.001]

Author’s response: Thank you for the advice on the missing literature.

https://doi.org/10.1007/s10072-021-05855-2 - Truijen et al. (2022) - this systematic review and meta-analysis does not mention the upper limb. We would like not to include it if you agree.

https://doi.org/10.1080/09638288.2020.1768301 - Marotta et al. (2020): this systematic review addressed Parkinson's disease and not stroke, focused on locomotion and not the upper limb, and did not focus on the home (apart from the inclusion of the study by Song et al. (2018)). This review does not seem to us to be the most suitable and we would like not to include it if you agree.

https://doi.org/10.1089/g4h.2012.0073 – Borghese et al. (2013): added, thanks. L.50

https://doi.org/10.1016/j.ijmedinf.2018.12.001- Chen et al. (2018): We have included this systematic review in the introduction at line 61, but indeed we could talk about it earlier in the introduction, we have reworked the review. Thank you.

Action: see changes in paragraph 1 and 2 highlighted

  1. [Comment: L52 from the point of view of evidence you need high intensity and high frequency, more than hard work ..]

Author’s response: Indeed, this sentence deserved to be detailed.

Action: see changes in l.68 highlighted

“Severity and dependence in ADLs may be associated with a decrease in the number of hours of activity of the affected UL and a decrease in activity ratios that should be considered when designing therapeutic interventions and setting goals [16]. In the recent literature, the intensity and frequency recommended for subacute stroke patients had been observed [17]. The authors reported that twice-daily exergaming, compared to high-intensity once-daily exergaming or lower-intensity once-daily standard care, produced superior effects on clinical and motor symptoms and quality of life in participants with subacute stroke [17]. These results are promising but it is not clear whether this is transferable to the home using remote monitoring and commercially available technology”.

  1. [Comment: 57 “In addition to maintaining functional improvements, telerehabilitation could monitor the follow up”]

Author’s response:

Action: see changes in l.44 highlighted

l.44 “In addition to maintaining functional improvements, telerehabilitation could monitor the follow-up [8] “.

Laver K, Adey‐Wakeling Z, Crotty M, Lannin N, George S, Sherrington C. Telerehabilitation services for stroke. Cochrane Database of Systematic Reviews 2020. https://doi.org/10.1002/14651858.CD010255.pub3.

  1. [Comment: 122 Risk of Bias is missing]

Author’s response: The risk of bias was assessed on the Physiotherapy Evidence Database (PEDro) scale in Table 2, l.149 and section l.365. Eight items evaluate the risk of bias (random allocation, concealed allocation, similarity at baseline, subject blinding, therapist blinding, assessor blinding, completeness of follow up, intention-to-treat analysis) and two items evaluate the completeness of statistical reporting (between-group statistical comparisons, and point measures and variability). We know that two instruments are commonly used to assess the risk of bias of trials of physiotherapy interventions: the Cochrane risk of bias (CROB) tool and the PEDro scale. With reference to Moseley et al. (2019), we could use either the CROB tool or the PEDro scale, as neither can be considered the gold standard for risk of bias evaluation.

Moseley AM, et al. Agreement between the Cochrane risk of bias tool and Physiotherapy Evidence Database (PEDro) scale: a meta-epidemiological study of randomized controlled trials of physical therapy interventions. PLoS One 2019;14(9):e0222770

Action: see changes in l.152 highlighted

“RCTs were evaluated using the Physiotherapy Evidence Database (PEDro) Scale, that generates a score out of 10 points, reflecting a study's internal validity, the methodological quality and potential risk of bias of each included study [26].”

  1. [Comment: 150 Did you only conduct MD or did you expect to use SMD? For what heterogeneity value did you conduct a random effect?]

Author’s response: We were forced to make changes due to errors. We performed an SMD in 2.1 and 3.1 because the objective assessments used different scales, we performed MD in 2.2 2.3 and 3.2 3.3 because the subjective assessments used the same scale. I2 > 50% was considered heterogeneous. Fixed and random effects models were used to pool study results with low and high heterogeneity, respectively.

Action: see changes in figure 2 and 3 highlighted

  1. [Comment: Figure 2 it’s non SMD .. it’s MD.. Is it the same outcome evaluated?]

Author’s response: We were forced to make changes due to errors. We performed an SMD in 2.1 and 3.1 because the objective assessments used different scales, we performed MD in 2.2 2.3 and 3.2 3.3 because the subjective assessments used the same scale

Action: see changes in figure 2 and 3 highlighted

  1. [Comment: 466 PROSPERO is missing]

Author’s response: Indeed, PROSPERO registration had not been done. We know that it was not mandatory but recommended. We accept your remark as a limitation of our systematic review.

  1. [Comment: 508 Again, I disagree on the term alternative .. there seems to be better and worse intervention, instead telerehabilitation can provide continuity, monitor and engage individuals and families.]

Author’s response: We agree that we have extrapolated the results using the term “alternative”. Then, we decided to use the word “support”. The exergame is one of the different modalities of telerehabilitation in the accompaniment around the person.

Action: see changes in l.531 highlighted

“This systematic review found that exergames in upper limb tele-rehabilitation seemed to be a relevant support for promoting post-stroke activity. Some situations may be favorable for remote rehabilitation interventions. The exergames should be included in the overall tele-rehabilitation ensure continuity, follow-up and participation of individuals and families”

Reviewer 4 Report

Dear Author(s)

1. Why do you write SMD below the figures, whereas you write MD in the figures and the text?

2. Thee are grammatical errors. "was independently" change to "were independently". etc.

3. The number of studies in each analysis is low.

4. Please add reference for "The heterogeneity of the results was reported as I2 followed by a percentage:"

5. Line 101: "non-randomized controlled trials" change to "non-RCTs". Line 149: "randomized controlled trial" change to "RCT", etc. Please pay attention to abbreviations and their full names.

6. Please add trial sequential analysis.

Author Response

  1. [Comment: Why do you write SMD below the figures, whereas you write MD in the figures and the text?]

Author’s response: We were forced to make changes due to errors. We performed an SMD in 2.1 and 3.1 because the objective assessments used different scales, we performed MD in 2.2 2.3 and 3.2 3.3 because the subjective assessments used the same scale

Action: see changes in figure 2 and 3 highlighted

  1. [Comment: Thee are grammatical errors. "was independently" change to "were independently". etc.]

Author’s response: completed

Action: see changes highlighted

  1. [Comment: The number of studies in each analysis is low.]

Author’s response: We agree with your comment. We considered the small number of studies at the limit of the review in l.484. This aspect limited the analysis of the results, which could have allowed for subgroups with a larger number of studies. This number of studies did not allow to test for publication bias (tests for funnel plot asymmetry)

  1. [Comment: Please add reference for "The heterogeneity of the results was reported as I2 followed by a percentage:"]

Author’s response: We take this oversight into account, thank you.

Higgins JPT, Thomas J, Chandler J, Cumpston M, Li T, Page MJ, Welch VA (editors). Cochrane Handbook for Systematic Reviews of Interventions. vol. Version 6.3 (updated February 2022). Cochrane. 2022.

We also corrected a mistake in the percentages “between 75-100% to considerable heterogeneity”

Action: see changes 90% for 100% and add reference [24] highlighted.

  1. [Comment: Line 101: "non-randomized controlled trials" change to "non-RCTs". Line 149: "randomized controlled trial" change to "RCT", etc. Please pay attention to abbreviations and their full names.]

Author’s response: completed

Action: see changes in l.102 - l.150 – UL in l.66 - ADLs l.433 - l.464  highlighted

  1. [Comment: Please add trial sequential analysis.]

Author’s response: The Trial Sequential Analysis would be relevant given the small number of studies and patients impacting on statistical power. Results from systematic reviews and meta-analysis may often increase the likelihood of over- or underestimation. We wanted to contact an expert in order to carry out this analysis accurately, which required time. We are therefore unable to present it to you within this timeframe, and we apologise for this.

Round 2

Reviewer 3 Report

My concerns have been addressed

Author Response

Thank you for your reply that your concerns have been addressed. Changes have been made to be in agreement.

Reviewer 4 Report

Dear Author (s)

1. Please add trial sequential analysis.

2. Please search the Web of Science database.

3. There are grammatical errors "After discharge from the hospital to home," to "After discharge from the hospital to the home,". "six-month retention of MAL and WMFT gains were found" to "six-month retention of MAL and WMFT gains was found". "on community reintegration, mobility and quality of life after stroke" to "on community reintegration, mobility, and quality of life after stroke". "that long-term effects" to "that the long-term effects", "As in Brouwer et al. (2018)," to "As in the study of Brouwer et al. (2018)," etc.

4. Please change "This review suggests" to "systematic review and meta-analysis".

5. Please change "This systematic review found" to "This systematic review and meta-analysis found". etc.

6. Please add a short definition of each outcome in the method section.

Author Response

We would like to thank you for your feedback.

[Moderate English changes required]

Reply: we made changes to your recommendations, thank you

[1. Please add trial sequential analysis.]

Reply: thank you for your suggestions for the Trial Sequential Analysis (TSA). We have looked into this analysis and have called in our methodological and statistical colleagues. No one knows or has been trained in this method. We are aware that this parameter may be relevant. However, based on the results of our work, the TSA will not change the conclusion. Finally, we stress that this endpoint is not yet routinely used in systematic reviews and is not requested in the PRISMA guidelines. We took the decision not to add it to the review and we hope you will understand. Thank you.

 [2. Please search the Web of Science database.]

Reply: The Cochrane Library Handbook recommends the use of at least MEDLINE, CENTRAL and EMBASE for identifying reports of randomized controlled trials.

Higgins JPT, Green S. Cochrane handbook for systematic reviews of interventions: The Cochrane Collaboration, London, United Kingdom. 2011

We used additional databases such as CINAHL and SCOPUS to be able to identify the literature related to the topic of interest adequately. CINAHL is the reference database for professionals in the paramedical field. Scopus is a multidisciplinary database of scientific journals in the fields of science, medicine, and technology. It was decided to focus on its two additional databases.

[3. There are grammatical errors "After discharge from the hospital to home," to "After discharge from the hospital to the home,". "six-month retention of MAL and WMFT gains were found" to "six-month retention of MAL and WMFT gains was found". "on community reintegration, mobility and quality of life after stroke" to "on community reintegration, mobility, and quality of life after stroke". "that long-term effects" to "that the long-term effects", "As in Brouwer et al. (2018)," to "As in the study of Brouwer et al. (2018)," etc.]

Reply: we made changes to your recommendations. Thank you

[4. Please change "This review suggests" to "systematic review and meta-analysis".]

Reply: we made changes to your recommendations. Thank you

[5. Please change "This systematic review found" to "This systematic review and meta-analysis found". etc.]

Reply: we made changes to your recommendations. Thank you

[6. Please add a short definition of each outcome in the method section.]

Reply: we detailed the two categories of assessment in section l164. Thank you for your advice